# Survivability of Anhydrobiotic Cyanobacteria in Salty Ice: Implications for the Habitability of Icy Worlds

**DOI:** 10.3390/life9040086

**Published:** 2019-11-22

**Authors:** Barbara Cosciotti, Amedeo Balbi, Alessandra Ceccarelli, Claudia Fagliarone, Elisabetta Mattei, Sebastian Emanuel Lauro, Federico Di Paolo, Elena Pettinelli, Daniela Billi

**Affiliations:** 1Department of Mathematics and Physics, University of Rome Tre, 00154 Rome, Italy; cosciotti@fis.uniroma3.it (B.C.); ceccarelli.ap@gmail.com (A.C.); mattei@fis.uniroma3.it (E.M.); lauro@fis.uniroma3.it (S.E.L.); dipaolo@fis.uniroma3.it (F.D.P.); pettinelli@fis.uniroma3.it (E.P.); 2Department of Physics, University of Rome Tor Vergata, 00133 Rome, Italy; amedeo.balbi@roma2.infn.it; 3Department of Biology, University of Rome Tor Vergata, 00133 Rome, Italy; claudia.fagliarone@hotmail.it

**Keywords:** vitrification, desert cyanobacteria, laboratory simulations, habitability, Europa, icy moons, liquid veins, ice crystals

## Abstract

Two anhydrobiotic strains of the cyanobacterium *Chroococcidiopsis*, namely CCMEE 029 and CCMEE 171, isolated from the Negev Desert in Israel and from the Dry Valleys in Antarctica, were exposed to salty-ice simulations. The aim of the experiment was to investigate the cyanobacterial capability to survive under sub-freezing temperatures in samples simulating the environment of icy worlds. The two strains were mixed with liquid solutions having sub-eutectic concentration of Na_2_SO_4_, MgSO_4_ and NaCl, then frozen down to different final temperatures (258 K, 233 K and 203 K) in various experimental runs. Both strains survived the exposure to 258 K in NaCl solution, probably as they migrated in the liquid veins between ice grain boundaries. However, they also survived at 258 K in Na_2_SO_4_ and MgSO_4_-salty-ice samples—that is, a temperature well below the eutectic temperature of the solutions, where liquid veins should not exist anymore. Moreover, both strains survived the exposure at 233 K in each salty-ice sample, with CCMEE 171 showing an enhanced survivability, whereas there were no survivors at 203 K. The survival limit at low temperature was further extended when both strains were exposed to 193 K as air-dried cells. The results suggest that vitrification might be a strategy for microbial life forms to survive in potentially habitable icy moons, for example in Europa’s icy crust. By entering a dried, frozen state, they could be transported from niches, which became non-habitable to new habitable ones, and possibly return to metabolic activity.

## 1. Introduction

Potentially habitable worlds are usually defined in terms of three basic requirements: the presence of liquid water, the availability of biogenic elements and a free energy source [1]. In our solar system, two icy worlds are currently attracting significant attention as environments which may meet all such criteria: Enceladus and Europa, moons of Saturn and Jupiter, respectively [2]. There is strong evidence that both moons have a large reservoir of liquid water below the icy crust, and that hydrothermal activity may take place on the ocean floor [3,4]. Hypothetical life-forms could be transported from the subsurface ocean with the water plumes [5] and deposited on or near the icy surface. Within the ice shell additional habitats might be present, resembling veins at grain boundaries and brine zones as reported for glaciers and ice sheets on Earth [3,4], albeit subsurface melt pockets due to tidal convection or eutectic melting [6]. Energy for metabolic processes could be provided on the seafloor by tidal activity, radiogenic heating and serpentinization [7]. 

Europa and Enceladus are astrobiologically relevant not only because they are potentially among the most interesting targets in the Solar System for finding life, but also because they can serve as candidates for a case study of icy planetary environments, that may be quite common in the universe and support life even outside the circumstellar habitable zone [8,9,10]. It is therefore crucial to complement the *in situ* and remote study of these icy moons with laboratory simulations in order to assess their habitability.

Cold-adapted microbes thrive in environments considered as analogues of potentially habitable icy worlds [11], where they occur in three microhabitats: (i) within briny veins at the triple grain-boundaries in polycrystalline ice [12,13]; (ii) associated with mineral grains in permafrost in thin films of unfrozen water [14]; and (iii) inside ice crystals [15]. 

In liquid veins, microbes need to cope with increased solute concentrations and have access to water and energy; however, when they are encased in solid ice, the limiting factor to their metabolism is nutrient diffusion through a solid, which is orders of magnitude slower than through a liquid [12]. Survival clearly depends on various physical-chemical features and cold-adaptation strategies. Ice-inhabitants accumulate compatible solutes to prevent osmotic shock, and synthetize ice-binding or anti-freeze proteins (localized on their cell surface or secreted outside) to prevent ice-crystal formation and cellular damage; they also release exopolymeric substances that may function as anti-freeze agents [16].

A key question is whether microbes simply survive by being trapped frozen in glacial ice, or if they are metabolically active, thus repairing macromolecular damage and even dividing and completing their life cycle. Despite the relevance of the issue, there is no generally accepted value for the lower temperature limit for life on Earth. Microbial metabolism has been reported, albeit with slow rates, occurring at temperatures in the range from 256 K to 253 K [17]. Indeed, evidence suggests that, during cooling in the presence of external ice, microbes will undergo freeze-induced desiccation and glass transition (vitrification). Thus vitrified, dormant cells are able to survive at very low temperatures [17]. Indeed, by extrapolating metabolic data at subzero temperatures, the threshold for microbial metabolism was set at 233 K [18].

The presence of salts, which lower the freezing point of the solution down to the eutectic temperature (or even lower if in metastable state), might expand the temperature range for potentially habitable extra-terrestrial environments. In polycrystalline ice, the existence of liquid veins depends on the ice temperature and on the eutectic temperature of the system (i.e. the temperature at which the solution precipitates as a mixture of solid salt and ice). In a binary system, according to equilibrium thermodynamics, the veins are liquid when the ice temperature is above the eutectic temperature of the system, whereas an immiscible solid mixture of ice and salt crystals freezes below such temperature (see [19,20] for details). In natural terrestrial ice many chemical species are present together, thus the existence and predominance of a microenvironment over the other (i.e. liquid veins or solid grains) depends on the type of chemical impurities and the temperature of the ice. 

On Europa’s surface, the presence of magnesium sulfate, sodium sulfate and sodium carbonate (and, potentially, of sulfuric acid) has been revealed by Galileo Near-Infrared Mapping Spectrometer [21]. Recently, the additional presence of sodium chloride was suggested by spectral analysis made with the Hubble Space Telescope [22]. On Earth, sodium chloride is the most common salt found in high-salt environments, although calcium chloride, sodium and magnesium sulfates have been reported in Don Juan Pond, Antarctica [23] and Spotted Lake, Canada [24], respectively.

The investigation of microbial survival in different solutes under freezing conditions is relevant to better define life-limiting factors and to evaluate the possibility of non-Earth icy environments being habitable [11]. Very little is known on the behavior of microbes in salt solutions under non-Earth conditions. For instance, bacteria have been reported to be viable and metabolically active in a type of ice not found on Earth, the so-called ice-VI, produced under high pressure conditions, and in which the solid phase of water, differently for the Earthly ice-Ih, is heavier than the liquid phase [25]. Hence laboratory icy-moon simulations might help to obtain new insights into bacterial endurance in extra-terrestrial cold environments. 

Here we sought to address the hypothesis that cyanobacteria from hot and cold deserts [26,27] remain viable after vitrifying upon exposure to different salt solutions at subfreezing temperatures. Indeed, the desiccation resistance of these non-akinete forming cyanobacteria depends on their ability to enter upon air-drying, an ametabolic state and recover their metabolism when water becomes available, a phenomenon known as anhydrobiosis [26,27]. Hence, we selected *Chroococcidiopsis* sp. CCMEE 171 isolated from endolithic communities in the McMurdo Dry Valleys in Antarctica, considered the coldest and driest place on Earth, where endoliths are metabolically active at 253 K [28] and persist in a dried-frozen state for most of the year [29]. On the other hand, *Chroococcidiopsis* sp. CCMEE 029 isolated from the Negev Desert, Israel, was selected due to its capability of surviving, when dried, temperatures as low as 248 K, under laboratory conditions [30]. 

In the present work, these two anhydrobiotic cyanobacteria were exposed to water solutions progressively cooled down to temperatures much lower than their eutectic temperatures. Since the salt concentration in icy moon crusts is unknown, a typical impurity load found in terrestrial glacial ice was used as a reference [12]. Thus, sub-eutectic solutions (10^−4^ M) of Na_2_SO_4_, MgSO_4_ and NaCl were prepared, and each solution was frozen down to three final temperatures, namely 258 K, 233 K and 203 K, in different laboratory simulations. After exposure, survival was evaluated by monitoring the growth capability after transfer to standard growth medium and incubation under routine conditions.

## 2. Materials and Methods

### 2.1. Cyanobacterial Strains and Culture Conditions

The strains used in our study are part of the Culture Collection of Microorganisms from Extreme Environments (CCMEE), maintained at the Department of Biology, University of Rome Tor Vergata. Hereafter, they are indicated as CCMEE 029 and CCMEE 171. *Chroococcidiopsis* sp. CCMEE 029 (N6904) and *Chroococcidiopsis* sp. CCMEE 171 (A789-2) were isolated by E. Imre Friedmann and Roseli Ocampo-Friedmann from cryptoendolithic growth in sandstones in Makhtesh Ramon, Negev desert (Israel) and in the University Valley, McMurdo Dry Valleys (Antarctica), respectively. For strain CCMEE 171 the growth optimum temperature is about 25 °C (unpublished), as reported for CCMEE 134, isolated from Beacon Valley in Antarctica [31]. The two strains are not axenic but routine transfer on agarized BG-11 medium [32] reduced the bacterial contamination to about 0.0001%. Cyanobacteria were grown under routine conditions at 25 °C in liquid BG-11 medium under a photon flux density of 40 μmol/m^2^/s provided by fluorescent cool-white bulbs under continuous light illumination. For the simulations reported below, cultures in the exponential growth phase (0.1–1 × 10^7^ cells/ml) were used.

### 2.2. Salty-Ice Environments

Icy samples were made, following [12], by preparing 10^−4^ M salt binary solutions, in ultrapure and sterile water (ddH_2_O), with MgSO_4_, Na_2_SO_4_ and NaCl as solute. For each salt solution three different simulations were performed at 258 K, 233 K and 203 K, by using a climatic chamber capable of reaching 198 K (Angelantoni DY340C, see Figure 2e,f in [33]). The temperature was measured by two Pt100 sensors, one located close to the sample and the other inside the chamber. The climatic chamber is hosted at the Mathematics and Physics Department of Roma Tre University and was designed and built to study different ice physical properties, as a support to the RIME experiment on board the JUICE 2022 ESA mission. It allows to control both temperature and humidity, and to generate temperature gradients as fast as 2 K/min, in cooling mode, and 4 K/min, in heating mode. In each experimental run the cooling time was different, about 20 min, 33 min and 48 min, at 258 K, 233 K and 203 K, as shown in Figure 1. For each experimental run, once reached the lowest value the sample was maintained at such temperature for 2 h. 

### 2.3. Exposure of Cyanobacteria to Salty-Icy Environments

Cyanobacteria were resuspended in 10^−4^ M solutions of MgSO_4_, Na_2_SO_4_ or NaCl according to [12] and exposed to the cooling procedure described above. Samples were prepared as follows: 1-mL culture aliquots were pelleted by centrifugation at 5000 *g* for 10 min, resuspended in 5 mL of each salt solution or in 5 mL of ddH_2_O, and transferred in 25 cm^2^ culture flasks. Controls were prepared from 1-mL culture aliquots resuspended in 5 mL of BG-11 medium and maintained at room temperature (RT). After each simulation, the icy samples were allowed to thaw at RT for about 2 h, washed with BG-11 medium, resuspended in 5 mL of BG-11 medium and incubated under routine culture conditions. 

### 2.4. Evaluation of the Effects of Salt Solutions and Ddh_2_o on Cell Viability

Cyanobacterial viability after incubation in salt solutions and in ddH_2_O at RT was evaluated as follows: 1-mL culture aliquots were resuspended in 5 mL of 10^−4^ M solutions of MgSO_4_, Na_2_SO_4_ and NaCl or ddH_2_O as described above, and incubated at RT for 4 h (in parallel to salty-ice simulation experiments). Then cells were washed with BG-11 medium, resuspended in 5 mL of BG-11 medium, transferred in 25 mL culture flasks and incubated under routine culture conditions.

### 2.5. Exposure of Air-Dried Cyanobacteria to 233 K and 193 K

Air-dried cyanobacteria were obtained from 1-mL culture aliquots by centrifugation at 5000 *g* for 10 min and dried overnight under a sterile air stream. Dried pellets were exposed as follows: (i) at 233 K for 2 h by using the climatic chamber; (ii) 193 K for 72 h by using a laboratory refrigerator. Frozen samples were allowed to thaw at RT for about 2 h, resuspended in 5 mL of BG-11 medium in 25 cm^2^ culture flasks, and incubated under routine conditions. Controls were prepared by air-drying 1-mL culture aliquots as reported above, and by resuspending the air-dried pellets in 5 mL of BG-11 medium at the same time of the frozen samples. 

### 2.6. Cell Viability

The growth rate of each exposed sample and control transferred in 5 mL of BG-11 medium under routine culture conditions, was monitored for 3 weeks by performing optical density measurements at 730 nm. Values are reported as means, based on two independent simulations with three replicates.

## 3. Results

### 3.1. Salty-Ice Environments

The conditions of the different experimental runs are summarized in Table 1. According to thermodynamics models of binary systems, because water and salts are totally miscible in the liquid phase but immiscible in the solid phase, below the eutectic temperature they should form a solid mixture of salt and ice grains. Thus, as in this study the salt solutions are quite diluted (10^−4^ mol/L), it is expected that far below the eutectic temperature the mixture is essentially composed by a matrix of polycrystalline ice with grains having an eutectic composition. Such grains are made of pure ice and pure salt crystals finely dispersed or distributed in alternate layers [19]. Thus, during cooling, the ice crystals segregate as a solid phase whereas the liquid phase becomes progressively more concentrated in salts. Once reached the eutectic temperature the salt solution starts to freeze. Table 1 shows that the eutectic temperatures of MgSO_4_ and Na_2_SO_4_ (i.e., 269.55 K and 271.99 K, respectively) are well above 258 K, while the eutectic temperature of NaCl (252.06 K) is lower than 258 K. Thus, it is expected that at 258 K the salty solutions containing MgSO_4_ and Na_2_SO_4_ are totally frozen, whereas the NaCl sample is only partially frozen, as it should contain a non-negligible quantity of salty liquid solution (probably in veins) having a concentration close to that of the eutectic composition (see Table 1). Conversely, because the other cooling experiments reached much lower final temperatures (233 K and 203 K), which are well below the eutectic temperatures of the three salt solutions (see Table 1), it is expected that no liquid salt solution would be left inside the icy sample. Therefore, at such low temperatures the samples should be made of a mixture of polycrystalline ice (the dominant phase) and disseminated grans of pure salt and icy crystals (frozen eutectic). 

### 3.2. Cold And Hot Desert Strains Survived in Salty-Icy Conditions at 258 K

After the exposure for 2 h at 258 K in icy samples containing NaCl, Na_2_SO_4_ or MgSO_4_, cells of strains CCMEE 171 and CCMEE 029 were transferred to liquid BG-11 medium under routine growth conditions. Strain CCMEE 171 survived in the three frozen salt solutions, as shown by their growth recovery (Figure 2A). However, after 21 days of growth, cell densities were lower than that of control, i.e., cells maintained in BG-11 medium under routine culture conditions (Figure 2A). By contrast, strain CCMEE 029 showed a growth recovery comparable to control after exposure to sub-freezing temperature in MgSO_4_ -H_2_O solution (Figure 2B)_;_ while a reduced growth occurred after exposure in the other salt solutions (Figure 2B). For both strains there were no survivors after exposure to 258 K in ddH_2_O (Figure 2A,B).

### 3.3. Enhanced Surival of the Cold Desert Strain in Salty-Icy Conditions at 233 K 

In the experiment where strain CCMEE 171 was exposed to 233 K for 2 h, growth recovery occurred after transfer to liquid BG-11 medium under routine culture conditions (Figure 3A), regardless of the salt solution (NaCl, Na_2_SO_4_ or MgSO_4_). However, after 21 days of growth, cell samples exposed to sub-frozen solutions of NaCl and Na_2_SO_4_, reached cell densities reduced if compared to control (Figure 3A). Conversely, strain CCMEE 029 exposed to 233 K in each one of the salt solutions showed a delay in the recovery and cell densities were lower than control (Figure 3B). For both strains there were no survivors after exposure to 233 K in ddH_2_O (Figure 3A,B).

### 3.4. Cold and Hot Desert Strains Died in Salty-Icy Conditions at 203 K 

No survivors occurred among cells of strains CCMEE 171 and CCMEE 029 after being exposed for 2 h to 203 K, regardless of the salt used, because no increase in cell densities was observed after transfer to liquid BG-11 medium under routine culture conditions (Figure 4A,B). 

### 3.5. Incubation in ddH_2_O and Salt Solutions Reduced Cell Viability 

The incubation of strains CCMEE 171 and CCMEE 029 in ddH_2_O and in 10^−4^ M solutions of NaCl, Na_2_SO_4,_ MgSO_4_ at RT for 4 h (in parallel to salty-ice simulation experiments) reduced cell viability as shown in Figure 4. After 21 days, strain CCMEE 171 showed cell densities reduced by the incubation in ddH_2_O, NaCl, Na_2_SO_4_, and MgSO_4_ to 87%, 71%, 66% and 70% of control (i.e. cells maintained in BG-11 medium under routine growth conditions), respectively (Figure 5). While cell densities of strain CCMEE 029 were reduced by the incubation in ddH_2_O, NaCl, Na_2_SO_4_, and MgSO_4_ to 98%, 77% 53% and 68% of control (i.e. cells maintained in liquid BG-11 medium), respectively (Figure 5).

### 3.6. Air-dried Cells of Hot and Cold Desert Strains Survived Sub-Freezing Temperatures 

Air-dried cells of strains CCMEE 171 and CCMEE 029 survived the exposure to 233 K for 2 h in all three salt solutions, as shown by the cell densities reached after 21 days of incubation in liquid BG-11 medium under routine culture conditions (Figure 6). The growth rate of strain CCMEE 171 was comparable to that of control (Figure 6A), whereas that of CCMEE 029 was slightly reduced (Figure 6B). When air-dried cells of strains CCMEE 171 and CCMEE 029 were incubated at 193 K for 72 h by using a laboratory refrigerator and transferred to liquid BG-11 medium under routine culture conditions, after 21 days their cell densities were comparable to control (not shown).

## 4. Discussion

We tested two anhydrobiotic cyanobacteria of the genus *Chroococcidiopsis* to evaluate their survivability in different icy environments, by using three salt solutions (NaCl, Na_2_SO_4_ and MgSO_4_) progressively cooled to different sub-freezing temperatures. In each experimental run the cooling time was different due to the cooling temperature gradient of 2 K/min, however the exposure time at 258 K, 233 K and 203 K was the same (2 h). Viability was assessed by monitoring the capability of entering cell division after transfer to routine growth medium and culture conditions, although the occurrence of viable-non-culturable cells should be taken into consideration. 

Both strains survived the freezing process at 258 K and 233 K, but died at 203 K. Moreover, strain CCMEE 171, isolated from the McMurdo Dry Valleys, Antarctica, showed an enhanced survivability compared to strain CCMEE 029, isolated form the Negev Desert, Israel. Indeed, strain CCMEE 171 is psychrotolerant (i.e., capable of metabolizing near 273 K and with optimal growth temperatures above 288 K [34]), and its cold-adaptation strategies might account for its enhanced survivability compared to strain CCMEE 029, under the performed icy simulations. Indeed, although adaptations of polar cyanobacteria are doubtless to be deciphered yet, mechanisms such as fatty acid desaturation or enzyme activity at low temperatures [35] might have contributed to its survival. On the other hand, the hot desert strain CCMEE 029, which is resistant to air-drying, showed the capability of facing freezing-induced desiccation.

The survival of both strains after exposure to 258 K in NaCl solution was expected, this temperature being slightly above the eutectic temperature of the NaCl-H_2_O system (252.35 K). Therefore, the sample occurred as polycrystalline ice with veins between ice grain boundaries filled with a salty solution with a salt concentration close to 20% (see Figure 3 in [19]). These conditions probably favored the microbial partitioning within the water-filled veins, since those have a large diameter (i.e. of the order of several µm) at 258 K [36]. The partitioning strongly depends on cell size; according to laboratory ice-formation simulations, microbes smaller than 2 µm would accommodate inside the veins, while larger ones (about 10 µm) would remain trapped within the ice crystals [12]. As other Pleurocapsalean cyanobacteria, strains CCMEE 171 and CCMEE 029 in addition to binary division that yields about 3 µm size single cells and four-cell aggregates, undergo multiple fission that occurs without increasing the mother-cell volume and producing smaller daughter cells [37]. These smaller cells could have fit into the liquid veins. However, both strains might have suffered the osmotic stress due to the increased salt concentration occurring in the salt solution filling the veins. In fact, desert cyanobacteria are adapted to face an external environment, namely the atmosphere, while the environment of cells under osmotic stress is an aqueous solution [38]. It is impossible to rule out that, under the simulations, the two cyanobacterial strains performed photosynthesis (if the light reaching the inside of the climatic chamber was enough) or exploited a light-independent energy generation pathway common to cyanobacteria [39] and reported for *Chroococcidiopsis* cells in deep subsurface rock samples [40]. In either case, their survivability might have been boosted by the exposure to low temperatures. Indeed, when exposed to eutectic NaCl solution *Planococcus halocryophilus* died within two weeks at room temperature; however, it survived at 277 K, an increased survival occurring at 258 K [41]. In addition, the maximum CaCl_2_ concentration suitable for its growth increased when the temperature was lowered from 298 K to 277 K [42].

Both strains CCMEE 171 and CCMEE 029 survived at 258 K also in Na_2_SO_4_ and MgSO_4_ salty ices. According to equilibrium chemistry, we should expect that the salty samples would be completely solid, however these results suggest that liquid salty solution could be present inside the icy sample in a supercooled metastable state [43]. Thus, large part of the cells could have partitioned in the vein microenvironment, even though we cannot exclude the entrapment of some of them inside ice crystals. The enhanced survival of strain CCMEE 029 in the presence of MgSO_4_, compared to CCMEE 171, is puzzling and it may be explained by taking into consideration that under this experimental condition extracellular polymeric substances might have created microscale salinity gradients, which affected the ice crystal growth, in a hitherto unpredictable way, thus increasing its habitability [44]. 

Even more surprisingly, CCMEE 171 and CCMEE 029 strains survived the exposure to 233 K in all three salty-ice samples, that is, to a temperature ranging between 20–30 K below the relevant eutectic temperatures. These results indicate that the metastable state could support the existence of the liquid salt eutectic solution inside the icy sample [43], preserving a favourable microenvironment for bacteria. It should be noticed that at these temperatures the dimension of the veins reduces sensibly, i.e. approximately 1µm at about 223 K [36]. In any case, we cannot exclude the possibility that cyanobacteria remained entrapped inside the ice crystals, even though so far bacteria have only been detected in veins and nodes [12]. 

In response to matric water stress, trehalose and sucrose are accumulated in strain CCMEE 029 [45] and strain CCMEE 171 (unpublished). These two non-reducing sugars act as compatible solutes [46] and are required during the initial stages of desiccation [38], although they also stabilize cells upon drying by replacing water molecules and allowing glass formation [47]. Moreover, trehalose not only provides protection during drying, but it is also an excellent cryoprotectant [48]. Indeed, a different accumulation of compatible solutes during freezing-induced desiccation might have accounted for differences in cyanobacterial survival.

In order to successfully vitrify during the exposure in salty ices, cyanobacteria should have accumulated trehalose and sucrose and should have entered the glassy state before intracellular ice formation. It has been reported that during a cooling of 20 K/min and in presence of external ice, microbes undergo freeze-induced desiccation and glass transition at temperatures between 263 K and 247 K [17]. In the present simulations, 2 K/min cooling rate was used: hence, the intracellular ice formation was probably avoided, and cells were probably dried before reaching the nucleation temperature [49]. 

Both strains CCMEE 171 and CCMEE 029 died at 203 K regardless of the salt used. It can be assumed that at this temperature the salt/water binary system is completely solid. Therefore, this value exceeded the survival temperature limit of these cyanobacteria, i.e. the temperature at which a cell enters vitrification but still resumes the metabolism when the temperature limit for metabolism is reached again [17]. 

Remarkably, air-dried cells of strains CCMEE 171 and CCMEE 029 survived the incubation for 3 days at 193 K (by using a laboratory refrigerator), as well as 2 h at 233 K (by using the climatic chamber). Hence, one can speculate that air-drying induced an adaptation different from the osmotic stress response, protecting the cells at subfreezing temperatures. Indeed, dried biofilms of *Sphingomonas desiccabilis* exposed to brines showed an enhanced resistance compared to hydrated biofilms, suggesting that drying before brining helps protect against brine exposure [50]. On the other hand, both strains CCMEE 171 and CCMEE 029 were killed by the exposure in pure water at 258 K, 233 K and 203 K. This was in line with the observation that freeze-thawing of *Planococcus halocryophilus* was less lethal in eutectic brines than in salt-free water [42].

In the present work, the endurance of air-dried cells at 193 K extended the survival temperature threshold of the two investigated cyanobacteria, thus broadening our expectations about the survival limits of life. Actually, as long as an organism can cope with freeze-thawing cycles, there is in principle no minimum low temperature for survival, because being in a frozen state is equivalent to being dormant [51]. 

Europa’s surface has temperatures ranging from 86 K to 132 K, hence far below the lowest limits known for life on Earth [17]. Nevertheless, the survival reported in this study is significant because it supports a highly speculative, but still intriguing role for vitrification in the context of the potential habitability of icy moons. On Europa, subduction events, due to a plate tectonic-like system, might force an upwards movement of microbial life forms from more favourable niches, such as the ice/water interface or the subsurface ocean, to near-surface temperatures where they would conceivably flash-freeze [52]. Therefore, vitrification might be a strategy for microbial life forms to survive in potentially habitable icy moons such as Europa’s icy crust. By entering a dried, frozen state, they could migrate from niches that became non-habitable to new habitable ones, and possibly return to metabolic activity. 

## Figures and Tables

**Figure 1 life-09-00086-f001:**
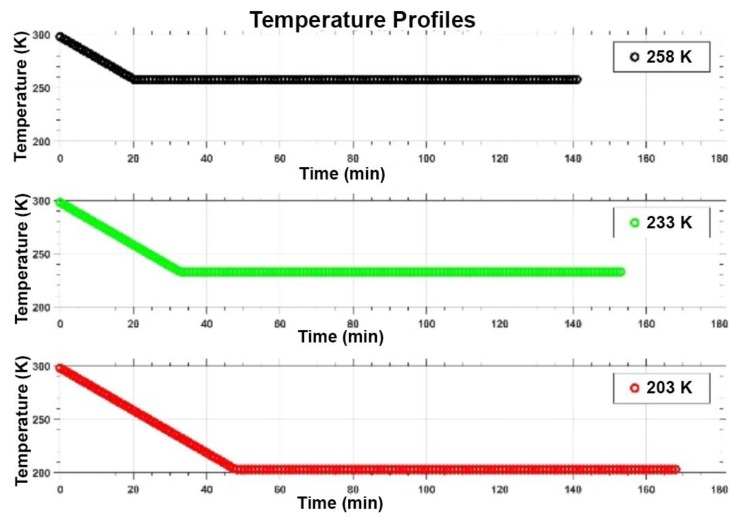
Temperature trend for the three experiments.

**Figure 2 life-09-00086-f002:**
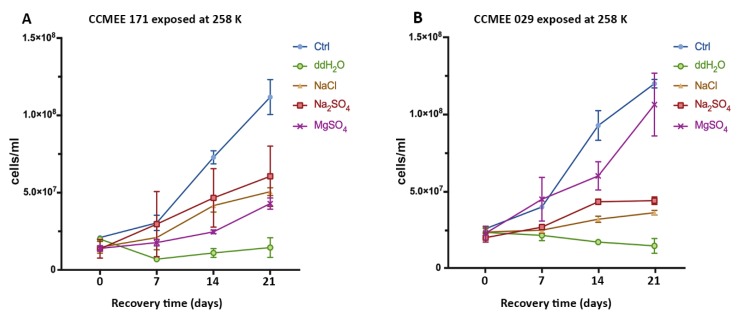
Growth curves of *Chroococcidiopsis* strains CCMEE 171 (**A**) and CCMEE 029 (**B**) transferred to liquid BG-11 medium under routine culture conditions after exposure to 258 K in icy samples containing NaCl, Na_2_SO_4_ or MgSO_4_. Ctrl: Control cells maintained in liquid BG-11 medium at RT during the simulation.

**Figure 3 life-09-00086-f003:**
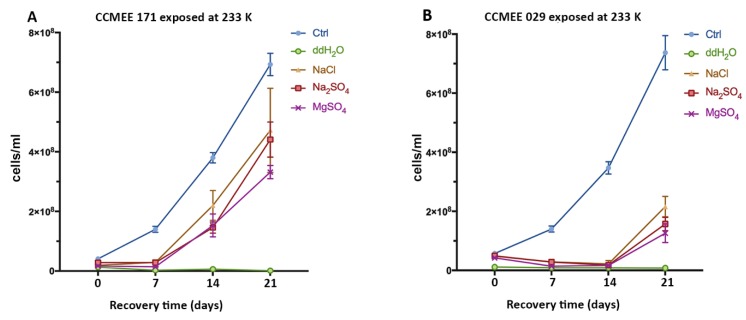
Growth curves of *Chroococcidiopsis* strains CCMEE 171 (**A**) and CCMEE 029 (**B**) transferred to liquid BG-11 medium under routine culture conditions after exposure to 233 K in icy samples containing NaCl, Na_2_SO_4_ or MgSO_4_. Ctrl: Control cells maintained in liquid BG-11 medium at RT during the simulation.

**Figure 4 life-09-00086-f004:**
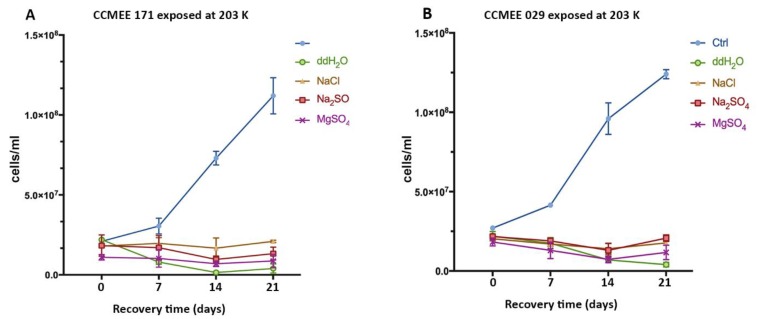
Growth curves of *Chroococcidiopsis* strains CCMEE 171 (**A**) and CCMEE 029 (**B**) transferred to liquid BG-11 medium under routine culture conditions after exposure to 203 K in icy samples containing NaCl, Na_2_SO_4_ or MgSO_4_. Ctrl: Control cells maintained in liquid BG-11 medium at RT during the simulation.

**Figure 5 life-09-00086-f005:**
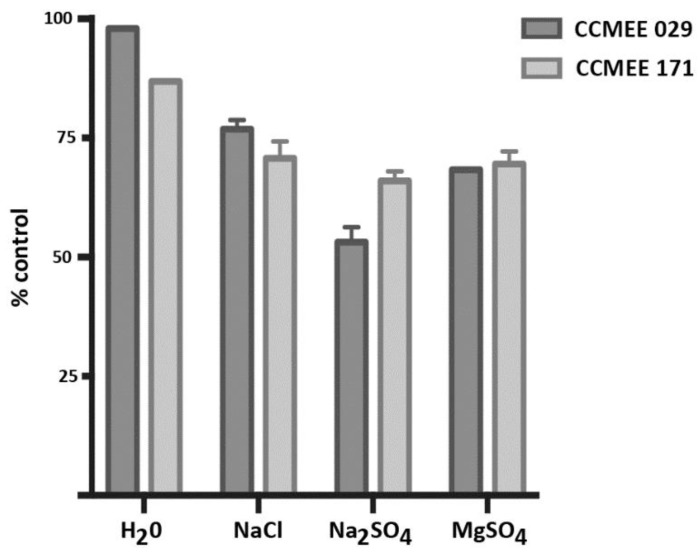
Cell densities of *Chroococcidiopsis* strains CCMEE 171 and CCMEE 029 incubated in ddH_2_O, NaCl, Na_2_SO_4_ or MgSO_4_ for 4 h at RT after incubation for 21 days in liquid BG-11 medium under routine culture conditions. Cell densities are shown as % of control (cells maintained in liquid BG-11 medium at RT during the treatment).

**Figure 6 life-09-00086-f006:**
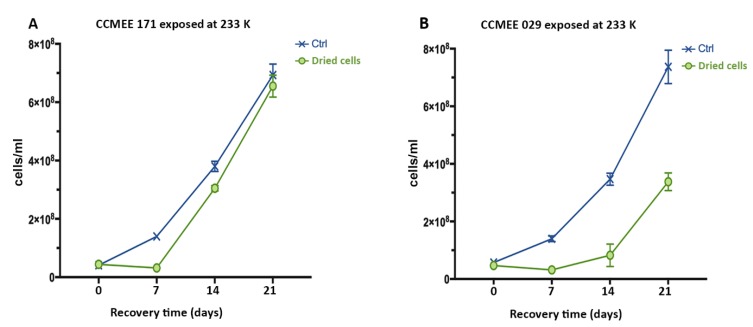
Growth curves of *Chroococcidiopsis* strains CCMEE 171 (**A**) and CCMEE 029 (**B**) exposed to 233 K as air-dried cells for 72 h and then resuspended in liquid BG-11 medium under routine culture conditions. Ctrl: Control air-dried pellets maintained at RT during the simulation and resuspended in liquid BG-11 medium.

**Table 1 life-09-00086-t001:** Experimental conditions and eutectic properties of the salty-ice samples used in this study.

Liquid Solution	Hydrated Form of Salt at Eutectic Concentration	Initial Salt Concentration (mol/L)	Eutectic Temperature/Concentration (K/wt%)	Sample Temperature (K)
**NaCl-H_2_O**	NaCl·2H_2_O	10^−4^	252.35/23.3	258
				233
				203
**Na_2_SO_4_-H_2_O**	Na_2_SO_4_·10 H_2_O	10^−4^	271.99/3.8	258
				233
				203
**MgSO_4_-H_2_O**	MgSO_4_·11 H_2_O	10^−4^	269.55/17.55	258
				233
				203
**ddH_2_O**	Pure ice	-	-	258
				233
				203

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
