# Peer review of "Survivability of Anhydrobiotic Cyanobacteria in Salty Ice: Implications for the Habitability of Icy Worlds"

_life, 2019, doi:10.3390/life9040086_

Round 1

Reviewer 1 Report

The manuscript “Survivability of anhydrobiotic cyanobacteria under water-ice/salty mixture simulations: Implications for the habitability of icy worlds” contains some interesting results that advance our understanding of the potential for life on Icy Worlds. However, I would not recommend the manuscript for publication without significant changes.

The manuscript is quite confusing as currently written and would benefit from significant proof-reading and editing to enhance the clarity. As it stands it is very difficult to understand the methodology of the research, which reduces the impact of the results.

Apart from the confusing writing and structure, I have one major conceptual issue with the study. The authors use salt solutions at incredibly low concentration. These solutions will be concentrated as they are frozen in the experiments described, but the specific concentration in the ice veins as they freeze will vary significantly depending on the interplay of the cooling and concentration enhancement. This could vary significantly between the different salts and potentially between technical replicates because of the stochastic nature of ice formation. However, there is no attempt to quantify these effects, which would significantly strengthen the study. If the authors were able to include some details about this concentration enhancement I think it would significantly improve the manuscript.

My other major issue is that as it stands I struggle to understand the sequence of experiments that the authors actually made. The methods described in section 2.2 are unclear and confusingly structured. It isn’t clear how many of experiment they completed (they state “three different experiments were performed reaching 258K, …”- Line 127 but it isn’t clear if that refers to the three different temperatures, or biological replicates) under each condition. I would recommend structuring the methods section more clearly to separate each experiment – the results section is clearer on this part. What are the different types of samples (i, ii, and iii)?  It seems like the i samples are a control, but as described they also contain a control? They don’t seem to correspond to the list of conditions described in Table 1. There is also no description of how the error bars were calculated. I would find it difficult to replicate the methods of the study, so the authors should make these much clearer.

More detailed comments below:

The authors use the phrase “water-ice/salty mixture” repeatedly (including in the title), but in my opinion this does not accurately describe the experiments, as they use salt solutions (not mixtures) and the ice is formed during the experiments rather than before. It would significantly simplify the manuscript to use the more straightforward “salt solutions” or “brines” (although the latter is still inaccurate due to the low concentrations in question.

There are a number of minor vocabulary and grammatical errors throughout the manuscript, which should be addressed after corrections. It would benefit from being proof-read by a native English speaker before re-submission.

There are numerous repetitions and redundant sentences. For example, on lines 99-102 the authors state which salt solution they used, then immediately state the same. Parts of the methods are repeated throughout the results section unnecessarily, and parts of the results section are repeated unnecessarily in the discussion. In my opinion the manuscript could be significantly condensed without losing any information.

As the authors use only CFU counts for quantification, they should note somewhere in the manuscript that this does not directly measure viability and leaves the potential for viable-non-culturable microbes in the samples.

Line 20 – I would not use ‘scored’ to describe survivors (also elsewhere in the manuscript) – “while there were no survivors” would be more than sufficient.

Line 24 – ‘migration’ implies active transport, rather than passive. I would use another term.

Line 63 – new sentence instead of a colon.

Line 66-68 – I would suggest a change to “In the presence of salts, which lower the freezing point of a liquid to the eutectic temperature of the salt/water mixture, might expand the temperature range for potentially habitable extra-terrestrial environments.”

Line 88 – I’m not sure ‘vitrification capability’ is a good term, surely the question is whether the cells can reanimate after vitrification, as they will vitrify whether or not they die?

Line 102-104 – Could the authors compare this concentration to what is expected on icy moons?

Line 135 – “saline solutions” where “salty solutions” was used before. I would simplify to “salt solutions”.

Line 137-139 – I don’t understand what this means, were the samples air dried before being put in the chamber, or while they were in the chamber? Why were they incubated in different refrigerators rather than the same conditions as the other samples? While the t

Line 142-143- There is no T-test data shown so I’m not sure why this is included.

Line 145-146 – Measurements are described as ‘optical density’ but plots shot CFU/ml – how were these cross-calibrated?

Line 150-153 – Repeat of methods.

Line 154-157 – Are there measurements of temperature through the experiments? It would be useful to see this profile over one or more of the experiments.

Table 1 – There’s a lot of repeated redundant information in the table, that could probably be removed (e.g. exposure time).

Line 160-162 – Repeat of methods. Not needed.

Line 167-168 – Plots show negative controls as > 0 CFU/ml, so presumably there was a minimum detection limit? Was the 258K experiment at this detection limit, or below it?

Figures – I would suggest a logarithmic y-axis perhaps.

Line 191-193 – Repeat of methods

Line 206 – Did these simulations include growth media?

Line 230 – The key here is not that the strains survive exposure to the salt solutions (as they are incredibly dilute and harmless) but that they survive the freezing process where the solution increases in concentration. I would make it much clearer in the discussion that this is the case.

Line 240 – I would say it’s entirely possible to rule this out, but it was not done in this experiment, and I’m not sure how this statement is relevant.

Line 246-248 – Why the change to celcius from kelvin here?

Line 251-256 – This paragraph is weak, the result for CCMEE 029 in MgSO4 at 258K is comparable to results for CCMEE 171 in the different salts at 233K, so why is only this one puzzling? This salt wouldn’t specifically induce an osmotic stress response compared to the others, and EPS wouldn’t protect against this salt in particular.

Line 270 – Celcius again instead of K

Line 273-276 – The measurements collected do not show cell death but loss of culturability.

Line 293-294 – The authors highlight the main finding of interest, regarding survival of vitrified bacteria in cold salty environments, but at no point was it confirmed in the experiments that the cells had actually vitrified – with no independent confirmation, can we be sure this actually happened, or is it assumed it happened from the environmental conditions?

Author Response

A: We do agree with the reviewer about low salt concentration and concentration variation during freezing. Therefor we have largely modified the manuscript to make it better organized and readable and to clarify several important points. Unfortunately, we do not have any possibility to evaluate the concentration enrichment during freezing, but we can only speculate on the basis of the binary system eutectic diagram. Moreover, the metastable state could maintain liquid the salty solutions well below the eutectic temperature, making the concentration evaluation even more difficult. We have addressed these aspects in different parts of the manuscript.

A: We re-wrote Materials and Methods in order to describe how many experiments were performed, and how controls were performed.

A:We have substituted the phrase “Water ice/salty mixture” by using different terminologies in the manuscript depending on the context. Mainly we used salty ice as it was changed in the title.

A (99-102). we fixed the introduction for the repeated part on salt solutions. Additional parts of the manuscript were re-written as requested.

A: we modified the discussion as requested: “although the occurrence of viable-non-culturable cells should be taken into consideration.“

A(L20): the term “scored” was removed from the manuscript as suggested.

A (L 24): the term “migration” was replaced by “could be transported”

A (L 63): A new sentence was added instead of the colon as requested .

A (L 66-68): The suggested sentence was added as suggested as follows: The presence of salts, which lower the freezing point of the solution down to the eutectic temperature, might expand the temperature range for potentially habitable extra-terrestrial environments.

A (L 88): the term “vitrification capability’ was modified as follows: “Here we sought to address the hypothesis that cyanobacteria from hot and cold deserts [26-27] remain viable after vitrifying upon exposure to water-ice/salty mixtures at subfreezing temperatures.”

A (L102-104): It has been suggested that the ocean below the icy crust should have a eutectic concentration; however, due to various proposed processes favouring the upwelling of the water, it is difficult to estimate the salt concentration in the liquid water close to the surface. For this reason, it will be important to run extensive experiments, starting from low concentration of salts up to supersaturated brines, to explore the entire range of possible habitats.

A (L135): the term “salt solutions” was used as requested.

A (L 137-139): Air-dried samples preparation is described in in the revised Materials and Methods, where the new section “2.4 Exposure of air-dried cyanobacteria at253 K, 233 Kand 193 K” was added. The reason of the two different treatments cells was to test a low temperature for a prolonged period of time, so we used lab refrigerators In order to make our main intent clearer we removed the incubation at 253 K and we reported only the result for the incubation at 193 K .

A (L 142-143): the sentence “significance assessed by using Student's t-test” was removed.

A (L 145-146): in the ‘optical density’ the term CFU/ml was replaced with cells/ml

A (L 150-153) the section  “3.1. Water-ice/salty mixture simulations” and Table 1 were moved to Materials and Methods.

A: (L 154-157) Measurements of temperature through the experiments were reported in fig. 1 of the revised manuscript.

A (Table 1) – Table was made less redundant and more useful information were added.

A (L 160-162) we deleted the following sentence as requested: “The system kept on cooling until eutectic temperature was reached, after which the remaining solution was frozen as a mixture of ice and salt.”

A (L 167-168): the negative control, e.g. cells in ddH20 showed decreasing absorbance because OD readings were reduced over time presumably due to cell lysis following cell death.  CFU/ml was also replaced by cells/ml because only turbidity was determined rather than colony forming capability.

A (Figures) we prefer the figure rending without using logarithmic y-axis.

A (L 191-193) :this was modified by adding the new figure 4 showing Growth curves after exposure to203 K in icy samples containing NaCl, Na2SO4or MgSO4.

A (L 206): the simulations were performed in order to evaluate the effects of the incubation in salt solutions and ddH2O on cell viability. Cells in liquid BG-11 medium were used as control. This was added in the re-written Material and methods.  Also the figure legend was fixed as follows: "Figure 5. Cell densities ofChroococcidiopsisstrains CCMEE 171 and CCMEE 029 incubated in ddH20, NaCl, Na2SO4or MgSO4for 4 hrsat RTafter incubation for 21 days in liquidBG-11 medium under routine culture conditions. Control: cells maintained in liquid BG-11 medium at RT during the treatment.

A (L 230): the following sentence was added as suggested “Both strains survived the freezing process at 258 K and 233 K, where the salt solution increased in concentration, but died at 203 K.”

A (L 240): we better explained this point as follows: As other Pleurocapsalean cyanobacteria, strains CCMEE 171 and CCMEE 029in addition to binary division that yields about 3 µm size single cells and four-cell aggregates, undergomultiple fission, that occurs without increasing the mother-cell volume, thus producing smaller daughter-cells [37]. These smaller cells could have fit into the liquid veins.

A (L 246-248): Celcius were replaced by Kelvin as requested.

A (L 251-256):Indeed it is not easy to explain this result, we can only speculate that “Although under this experimental condition extracellular polymeric substances might have created microscale salinity gradients, which affect ice crystal growth, in a hitherto unpredictable way, thus increasing its habitability [44]. “

A (L 270) Celcius replaced by Kelvin

A (L 273-276) We adeded the suggested sentence “although the occurence of viable-non-culturable cells should be taken into consideration

A (L 293-294): we asssume vitrification to occur on the basis of the capability of the two strains to accumulate trehalose and sucrose and the experimental conditions. This is reported in the discussion as follows: “In the over all, the scored survival should be correlated to the cyanobacterial capability of entering a state of vitrification under the laboratory icy-moon simulations.”

Reviewer 2 Report

The ms "Survivability of anhydrobiotic cyanobacteria under water-ice/salty mixture simulations: Implications for the habitability of icy worlds" by Cosciotti et al. is an interesting piece of work with interesting results. It will convenient if the authors improve the ms clarifying the following questions: is not clear why authors choose the concentration of salts used in the experiment(10-4M). The description of the methodology should be improved, it is difficult for non experts to follow the procedures: e.g. what was the cell density of the cyanobacteria cultures used, how samples were thawed after exposition to different temperatures, what were the optimal growth conditions used, how air-dried cell pellets were obtained, what is the composition of BG-11 medium.  

Author Response

A: To clarify why we chose 10-M salt solutions we added in the paragraph :

“In the present work these two anhydrobiotic cyanobacteria were exposed to water solutions progressively cooled down to temperatures much lower than their eutectic temperatures. Since the salt concentration in the icy-moon crusts is unknown, a typical impurity load found in terrestrial glacial ice was used as a reference [12]. Thus, sub-eutectic solutions (10-4 M) of Na2SO4, MgSO4 and NaCl where prepared, and each solution was frozen down to three final temperatures, namely 258 K, 233 K and 203 K, during different laboratory simulations”.

Indeed, the objectives of the paper was to start examining the potential habitats for life on Europa and the lower limits for biological activity with respect to temperature and salinity. Thus, this paper is a first step of a series of experiments devoted to study the ice environment as possible habitat. The sub-eutectic condition implies that during cooling the liquid phase enriches in salts up to the eutectic concentration and eutectic temperature. Theoretically, below such temperature the solution should be completely solid, however a metastable state could exist below the eutectic temperature. This condition is difficult to be determined and properly studied in laboratory.

We addressed the other points in Materials and Methods as follows:

-cultures in the exponential growth phase (0.1 – 1.5 x 107 cells/ml) were used the icy samples were allowed to thaw at RT for about 2 hrs, washed with BG-11 medium, resuspended in 5 ml of BG-11 medium and incubated under routine culture conditions. Cyanobacteria were grown under routine conditions at 25 °C in BG-11 liquid medium [32] under a photon flux density of 40 μmol/m2/s provided by fluorescent cool-white bulbs under continuous light illumination. Air-dried cyanobacteria were obtained from 1-ml culture aliquotes bycentrifugation at 5000 g for 10 minutes and overnight drying under a sterile air stream. The composition of BG-11 medium was given in ref [32].

Reviewer 3 Report

The manuscript of Cosciotti et al. describes the viability of two Chroococcidiopsis strains under freezing temperatures and concentrations of different salts. The manuscript is interesting, but there are major considerations to be published. Introduction is well-written and describes the context of the experiments. Material and methods are, however, confusing and lack important information (see comments per line below).  It is difficult to relate the experimental setup with the corresponding results and figures, and this must be fixed. One of the major concerns is that the authors could not control how long the cells were in liquid veins (which is the most relevant part from an astrobiological point of view). Moreover, the manuscript must be carefully check for correct spelling. Therefore, I would not recommend the publication of the manuscript in the present form.

Comments per line in the text:

L26: please, change to “desert cyanobacteria”

L54: please, add "to prevent osmotic shock" (or similar explanation) after “accumulate compatible solutes” for a better understanding.

L55: please, add “to prevent ice-crystal formation and cellular damage” (or similar explanation) after “anti-freeze proteins” or “secreted outside”.

L64: finished the sentence “at a very low temperatures”

L64-65: this sentence and L60-61 are confusing since both explain similar things. Please, rewrite for better understanding.

L107: are cyanobacterial cultures unispecies or only unialgal? Please, specify it.

L109: remove “cryptoendolithic in nubian sandstone” or fix the sentence somehow.

L111: add “respectively” after “(CCMEE)”.

L113: remove one “at”

L128: please, be consistent with the unit of Temperature (ºC or K). K is fully correct if the authors prefer to use it. The use ºC is, however, widely spread in biological studies and it would help readers to readily compare the temperatures used in this study with temperatures used in other studies.

L131-139: I do not understand the difference and the scientific point behind the two types of samples explained in i) and ii) for this experiment. What do you mean with room conditions? Room temperature? Moreover, I do not understand the reason for the two different treatments with air-dried cells (incubated for 2 h at 233 K in the climatic chamber or incubated for 3 days at 253 K and 193 K in refrigerators). Moreover, which are the conditions to air-dry the cells? In general, this section of Materials and Methods should be rewritten and complete for a better understanding of the experimental setup.

L145-147: Please, specify the volume you transferred of each sample experiment to the BG11 medium. Please, specify that is liquid medium (if so). Why authors use measurements at 730 nm instead of 665 nm for chlorophyll? 730 nm is close to infrared, which would measure turbidity. If culture strains are unialgal but not unispecies the measurement at 730 nm would take into account common heterotrophic bacteria that grows with cyanobacteria.

L150: this explanation and table fits more in Materials and Methods.

L151: If microbial live in brine veins are astrobiologically relevant, why the experimental design did not consider other salts concentrations to obtain ice and brines for sodium sulphate and magnesium sulphate?

L162: “survivors” is not the most correct word. Please, replace by “Strain CCMEE 171 survived after the exposure to…” or similar.

L188: replace “in fact” by “because”

L190: It is difficult to understand this section. Which type of samples correspond to this section? If they are from the sample experiment as before (with different temperatures) this would be repetition of results. Moreover, specify the media and routine conditions (L192).

L225: it would be interesting to include few lines of these possible different adaptation strategies for these strains. This would enrich this part.

L235-237: according to the previous sentence in which only 2-µm size cells would be fitted into the veins, it would be expected that 3-µm cells and aggregates were trapped in the ice. A deep explanation would be required.

L246: please, correct the spelling.

L261: This explanation is correct, but does not explain the possible difference in the viability between both strains. Are they accumulating different concentrations of compatible solutes during vitrification?

L266-272: The idea of this paragraph is the same as the previous one. Please, combine both paragraphs.

Figures 1 and 2: what is control in blue color? It has not been explained in the materials and methods. If this is the control of growth and has not been exposed to any of the temperatures, it must be also specified in the figure caption. Please, specify also that is liquid medium and the concentration of the salts. It applied to all figures in the manuscript. Moreover, growth experiments at 203K must be also shown either in a figure or in an explanation in the text showing the CFU of the control.

Author Response

A: The manuscript was fixed in order to make clear the experimental setup with the corresponding results and figures and checked for correct spelling.

A: As shown in the new Figure 1 the temperature profiles show that the cells were in the simulated icy conditions for two hrs: this period might be relevant for transportation from niches which became non-habitable to new habitable ones, and possibly return to metabolic activity.

A (L26): Desert cyanobacteria was fixed.

A (L54): to prevent osmotic shock was added.

A (L55): to prevent ice-crystal formation and cellular damage was added.

A (L64): The sentence “at a very low temperatures was fixed.

A (L64-65 and L60-61): Sentences L64-65: and L60-61 were fixed.

A (L107): The two cyanobacterial strains are not axenic, but bacterial contamination is very low as reported in Materials and Methods: “The two strains are not axenic but after routine transfer on agarized BG.11 medium reduced the bacterial contamination to about 0.0001%.”.

A L109): “cryptoendolithic in nubian sandstone” was fixed.

A (L111): respectively was added.

A (L128): Temperature were changed from °C to K) as requested.

A (L131-139): Room conditions was replaced by room temperature. The reason for the treatment with air-dried cells was to test at low temperature for a prolonged period of time. In order to make this intent clear we removed the incubation at 253 K and we presented only the result for the incubation at 193 K. The air-drying method was explained in the re-written Materials and Methods section.

A (L145-147): Each sample was transferred in 5-ml of BG-11 medium as reported in the re-written Materials and Methods section. Indeed we measured turbidity by using a wavelength were there is no pigment absorption as suggested for cyanobacteria. The used cultures are not axenic, although the bacterial contamination is very low as reported in the comment addressed above.

A (L150): table 1 was moved to Materials and Methods

A (L151): We have answered to the point on “If microbial live in brine veins are astrobiologically relevant, why the experimental design did not consider other salts concentrations to obtain ice and brines for sodium sulphate and magnesium sulphate? As follows: “According to thermodynamics models of binary systems, because water and salts are totally miscible in the liquid phase but immiscible in the solid phase, below the eutectic temperature, they should form a solid mixture of grains of salt and ice. Thus, as in this study the salt solutions are quite diluted (10-4 mol/l,), it is expected that far below the eutectic temperature the mixture is essentially composed by a matrix of polycrystalline ice with grains having a eutectic composition. Such grains are made of pure ice and pure salts crystals finely dispersed or distributed in alternate layers (e.g., -ref 35). Thus, during cooling, the ice crystals segregate as a solid phase whereas the liquid phase becomes progressively more concentrated in salts. Once reached the eutectic temperature the salty solution starts to freeze”.

We also would like to highlight that salt-water solution is a binary eutectic system; thus starting from a over-eutectic concentration and lowering the temperature would have had the effect of salt precipitation (as solid phase) and solution dilution. Below the eutectic temperature, ice and salts would have started to form and would have generated a solid matrix made of salt crystals, with grains of ice and salts disseminated in the matrix.

A (162): Survivors was replaced by “Strain CCMEE 171 survived”

A (L188): in fact” was replaced by “because” as requested

A (L 190:) The section was better explained by adding a new figure showing the results of the experiment at 203 K as requested by the reviewer’s last comment “Moreover, growth experiments at 203K must be also shown either in a figure or in an explanation in the text showing the CFU of the control.

A (L225): the following was added into the discussion as follows: “Indeed, although adaptationsof polar cyanobacteria are doubtless to be deciphered yet, mechanism such as fatty acid desaturation or enzyme activity at low temperatures [35]might have contributed to its survival. On the other hand, the hot desert strain CCMEE 029, being able to air-drying, showed the capability of facing desiccation due to freezing.”

A (L235-237) We added a deeper examination as follows: “Indeed, in addition to binary division that yields about 3-µm size single cells and four-celled aggregates,Chroococcidiopsisstrains undergomultiple fission, that occurs without increasing the mother-cell volume, thus producing smaller daughter cells [36]. These smaller cells could have fit into the liquid veins.”

A (L246): the spelling of cyanobacterial was corrected.

A (L261): The suggetsed possibility that the strains accumaluted different amount of compatibel solutes was added as follows: “Indeed, a different accumulation of compatible solutes during desiccation induced by freezing might account for the cyanobacterial different survival.”

A (L266-272): paragraphs with the same ideas were merged.

A (Figures 1 and 2): The legends were re-written that control specified. control were also better explained in the materials and methods. The result of growth experiment after exposure at 203 K is shown in Figure 4.

Round 2

Reviewer 1 Report

The manuscript is much improved in this version. The methods are much more readable and I feel like I could replicate the study now.

I noticed a few remaining (and new) minor errors in the text, and so I believe the paper would benefit from another pass of proof reading, but otherwise I would recommend the manuscript for publication.

Line 189 change to "as it should contain a non-negligible"

Line 245 change to "and MgSo4 to 98%, 77%, 53% and 68% of control" - no 'about' necessary

Line 274 "where the salt solution increased in concentration" - it's not clear what exactly this is referring to, as the solution concentration would increase in the other cases as well

line 281 change to "mechanisms"

Line 282 change to "desert strain CCMEE029, which is resistant to air-drying"

Line 283 change to "facing freezing induced desiccation"

Line 313-315 this sentence doesn't quite scan correctly and I'm not sure what the authors mean. Clarify this.

Line 324 - hanging sentence left from an edit it looks like

Author Response

A(L189) “contains” was changed into “contain”

A(L245) “about” was removed as suggested.

A(L274) the sentence “where the salt solution increased in concentration” was removed.

A(L281) “mechanism" was changed to mechanisms

A(L282) “which is resistant to air-drying” was used as suggested

A(L283) facing freezing induced desiccation was used as requested

A(313-315): the sentence was fixed as folloes “The enhanced survival of strainCCMEE 029 in the presence of MgSO4, compared to CCMEE 171, is puzzlingand it may be explained by taking into consideration that under this experimental condition extracellular polymeric substances might have created microscale salinity gradients, which affected the ice crystal growth, in a hitherto unpredictable way, thus increasing its habitability [44].”

A(L324): the hanging sentence was removed.

Reviewer 3 Report

The manuscript of Cosciotti et al. has substantially improved, with a clearer Materials and Methods section and Figures more readable. I recommend the publication in Life after some minor changes. In general, authors must pay attention to spelling.

L106: please, change to “were prepared”

L156: sentence suggestion “Cyanobacterial viability after incubation in salt solutions and ddH2O at room temperature was evaluated as follows:”  

L158-159: sentence suggestion “and incubated at RT for 3 hours (in parallel with salty-ice simulation experiments)”. Please, revise if it is 3 or 4 hours (L240)

L214: Please, fix with “Enhanced”

L240: sentence suggestion "(performed in parallel with salty-ice simulations)"

L248-251, Figure 5: Controls are not showed in this graph but % is related to cell concentration in control samples. Therefore, a small correction should be performed in this figure caption.

L324: Please, fix this sentence.

Author Response

A(L106): where was replaced by were prepared”

A(Ll56): the suggested sentence was used “Cyanobacterial viability after incubation in salt solutions and ddH2O at room temperature was evaluated as follows:”

A(L158-159): the suggested sentence was used “and incubated at RT for 3 hours (in parallel with salty-ice simulation experiments).

A(240): 4 hrs was revised.

A(L214): “Enhanced” was fixed.

A(L240): the suggested sentence was used “"(performed in parallel with salty-ice simulations)"

A (L248-251): this was fixed as follows:  Cell densities are shown as % of control.

A (L324): the hanging sentence was removed.